# Problematic Internet Use: Measurement and Structural Invariance Across Sex and Academic Year Cohorts

**DOI:** 10.3390/ejihpe15080145

**Published:** 2025-07-22

**Authors:** Mateo Pérez-Wiesner, Kora-Mareen Bühler, Jose Antonio López-Moreno

**Affiliations:** 1Department of Psychobiology and Methodology in Behavioral Sciences, Faculty of Psychology, Somosaguas Campus, Complutense University of Madrid, 28223 Madrid, Spain; kobuhler@ucm.es (K.-M.B.); jalopezm@ucm.es (J.A.L.-M.); 2Department of Psychology, Faculty of Health Sciences—HM Hospitals, University Camilo José Cela, 28692 Madrid, Spain; 3HM Hospitals, Health Research Institute, 28015 Madrid, Spain; 4MIDELOY Research-Madrid, 28922 Madrid, Spain

**Keywords:** emotion regulation, problematic digital media use, measurement invariance, structural equation modeling, digital media consequences, adolescents

## Abstract

The extensive use of digital media among adolescents has raised concerns about its impact on emotional development and mental health. Understanding the psychological factors behind problematic digital media use is essential for effective prevention. This study examined whether the relationships between emotion regulation (negative and positive), compulsive use, cognitive preoccupation, and negative outcomes linked to digital media are consistent across sex and academic year. We used a cross-sectional design with 2357 adolescents (12–16 years old) from Compulsory Secondary Education. Participants completed validated self-report questionnaires assessing problematic digital media use, and associated consequences in four domains: internet, video games, social networking, and messaging. Four structural equation models (SEMs), each focused on a media type, tested whether these relationships remained stable across sex and academic year. All models showed good fit, and differences between groups were minimal, supporting valid comparisons. Results confirm that emotion regulation difficulties and problematic digital media use are consistently associated with negative outcomes in all adolescents, regardless of sex or academic level. Preventive strategies targeting emotional regulation and digital media behaviors may be broadly applied to reduce emotional and functional problems related to excessive media use.

## 1. Introduction

The rapid growth of digital media, such as video games, internet browsing, social networking, and messaging, has raised concerns about their negative impact on mental health and social adjustment. Excessive use of smartphones and digital media among adolescents and young adults is linked to health issues, including depression, anxiety, sleep disturbances, and reduced cognitive functioning ([19]). Recent systematic reviews suggest that around 25% of children and adolescents display problematic smartphone behaviors similar to behavioral addictions, strongly associated with mood and sleep disorders ([17]).

A key factor behind these harmful patterns is difficulty in emotion regulation. Individuals who struggle to manage negative emotions frequently use digital media to cope, increasing their risk of problematic usage ([7]). Neuropsychiatric studies also show that impaired cognitive emotion regulation mediates the relationship between internet-use disorders and psychosocial issues ([14]). In young people, insecure parental attachments and maladaptive emotion regulation strategies increase vulnerability to problematic internet use ([13]).

Beyond general internet and smartphone addiction, problematic social media use correlates with narcissistic traits and low self-esteem. This suggests that social networks uniquely reinforce maladaptive emotional and self-regulatory behaviors ([2]). Meta-analytic studies highlight a bidirectional relationship between loneliness and problematic internet behaviors, illustrating the complex link between emotional well-being and digital media habits ([20]).

To understand these relationships, structural equation models (SEMs) have linked dimensions of emotion regulation (negative and positive), indicators of problematic use (compulsive use, cognitive preoccupation), and negative outcomes. For valid demographic comparisons, SEMs must demonstrate measurement invariance. This means ensuring the latent constructs and their relationships are stable and comparable across different groups ([5]).

In this study, we extend previous research by testing the measurement invariance of integrative SEMs across sex (male vs. female) and academic year (first to fourth year) for four types of digital media: video games, internet browsing, social networking, and messaging. Establishing measurement invariance is crucial to confirm that differences in digital media use and related outcomes across demographic groups reflect genuine variations, not measurement artifacts. Confirming invariance enables meaningful comparisons of emotional regulation, problematic usage patterns, and negative consequences among adolescents.

## 2. Materials and Methods

### 2.1. Participants

The sample consisted of 2357 adolescents enrolled in Compulsory Secondary Education (ESO) from the Autonomous Community of Madrid. Participants were recruited from 29 private and state-subsidized schools. Of the participants, 1087 were female (mean age = 13.71, SD = 2.62) and 1270 were male (mean age = 13.70, SD = 1.77). Distribution by academic year was: first year, 720 students (30.5%); second year, 694 students (29.4%); third year, 515 students (21.8%); and fourth year, 428 students (18.2%) (Table 1). Notably, given the narrow age range and the shared developmental stage of the participants, academic year was used as the grouping variable instead of chronological age, as it offers a more ecologically valid framework for examining peer-group influences and digital media use patterns.

### 2.2. Instruments

Problematic digital media use was assessed using the Questionnaire of Maladaptive Behavior toward ICT ([16]). This instrument includes 14 items rated on a six-point Likert scale (1 = strongly disagree, 6 = strongly agree) and evaluates four subscales: positive mood regulation (α = 0.83), negative emotion regulation (α = 0.73), compulsive use (α = 0.76), and cognitive preoccupation (α = 0.72). Each subscale is composed of a different number of items: four for positive mood regulation (score range: 4–24), three for negative emotion regulation (score range: 3–18), three for compulsive use (score range: 3–18), and four for cognitive preoccupation (score range: 4–24). Examples of items include: “When I am really happy, I use the internet to communicate it” (positive mood regulation), “When I feel bad, I use the Internet to feel better” (negative emotion regulation), “I find it difficult to control my Internet use” (compulsive use), and “I often worry about connecting to the Internet” (cognitive preoccupation). Negative outcomes related to digital media use were measured using the Questionnaire on Negative Consequences of ICT ([15]). This questionnaire assesses four domains of digital media activity: internet browsing, video games, social networking, and messaging. Each subscale includes seven items, resulting in a total of 28 items, all rated on the same six-point Likert scale. Cronbach’s alpha values were 0.77 for internet browsing, 0.86 for video games, 0.87 for social networking, and 0.89 for messaging. Each subscale has a minimum possible score of 7 and a maximum of 42. Examples of items include: “I spend my time browsing the internet when I should be doing other things” (internet browsing), “I have lost hours of sleep because of video games” (video games), “I have tried unsuccessfully to reduce the amount of time I spend using social media” (social networking), and “My academic or social activity is affected because I spend too much time on WhatsApp, SMS, etc.” (messaging). The instructions provided for each subscale include clarifications to distinguish between domains that may overlap in practice. For example, the section on social networking specifies platforms such as Facebook, TikTok, Instagram, and YouTube, while the messaging subscale refers explicitly to tools like WhatsApp, Messenger, Line, or SMS.

The construction of this second questionnaire was based on the main groups of diagnostic criteria from the DSM-5 for substance use disorder, including risky use, impaired control, social impairment, and withdrawal. These criteria were adapted to assess negative consequences in each digital activity domain, allowing for a nuanced understanding of problematic use depending on the specific online context, and were informed by a review of instruments previously published or adapted in Spain ([11]).

### 2.3. Procedure

We initially contacted 31 schools in Madrid (Spain); 29 agreed to participate. Before data collection, parents’ associations and school guidance departments were informed about the study’s aims, the questionnaires, and participant anonymity. Signed informed consent was obtained from parents, and assent from the adolescents. Participants completed questionnaires anonymously using tablets during regular class hours under teacher supervision throughout the 2022–2023 academic year. The study was approved by the Ethics Committee of Complutense University of Madrid.

### 2.4. Data Analysis

We conducted analyses using SPSS (Version 29) and R (Version 4.x) with the lavaan package for SEMs and semPlot for path diagrams. We managed missing item-level data (less than 3% per scale) using full information maximum likelihood (FIML). Model parameters were estimated using robust maximum likelihood (MLR) to account for non-normality.

For each digital media domain (internet, video games, social networking, messaging), we specified a structural equation model (SEM). Each SEM included four latent predictors (negative emotion regulation, positive emotion regulation, compulsive use, cognitive preoccupation) regressed on a single latent outcome (negative consequences). Covariances among predictors were freely estimated.

Measurement invariance was tested via multigroup analyses across sex (male vs. female) and academic year (first to fourth). We tested three nested models sequentially:

(a) Configural invariance: no constraints on loadings or intercepts. (b) Metric invariance: factor loadings constrained equal. And (c) Scalar invariance: intercepts are additionally constrained to be equal.

Model fit was assessed using the χ^2^ statistic, Comparative Fit Index (CFI), Tucker–Lewis Index (TLI), Root Mean Square Error of Approximation (RMSEA), and Standardized Root Mean Square Residual (SRMR). Accepted thresholds were CFI/TLI ≥ 0.95, RMSEA ≤ 0.06, and SRMR ≤ 0.08. We considered incremental fit changes (ΔCFI ≤ 0.01; ΔRMSEA ≤ 0.015; ΔSRMR ≤ 0.03) more important than χ^2^ differences due to sample size sensitivity. All statistical procedures complied with the transparency and reproducibility guidelines of the European Journal of Investigation in Health, Psychology and Education.

## 3. Results

First, we performed descriptive analyses to examine the central tendency and dispersion of the study variables, as well as their associations. Table 2 presents the means and standard deviations for each dimension of problematic digital media use and the reported negative consequences across the four digital media domains. Table 3 also displays the Pearson correlation matrix between these variables, revealing statistically significant and positive associations throughout. Among the dimensions, positive mood regulation showed the strongest correlations with the different media domains.

We conducted analyses of variance (ANOVA) to examine differences in digital media use across academic years and between sexes. For each digital media domain (internet browsing, video games, social networking, and messaging), a separate two-way ANOVA was performed, with academic year and sex as between-subject factors.

### 3.1. Internet Use

Significant main effects were found for academic year, F(3, 2349) = 10.04, *p* < 0.001, η^2^_p_ = 0.013, and sex, F(1, 2349) = 5.68, *p* = 0.017, η^2^_p_ = 0.002. There was no significant interaction effect between academic year and sex, F(3, 2349) = 0.55, *p* = 0.647, η^2^_p_ = 0.001. Male students reported higher internet use compared to female students, with an incremental increase observed across academic years for both groups.

### 3.2. Video Game Use

A significant main effect of sex emerged, F(1, 2349) = 267.49, *p* < 0.001, η^2^_p_ = 0.002, demonstrating considerably higher video game use among males. The interaction between academic year and sex was also significant, F(3, 2349) = 3.08, *p* = 0.026, η^2^_p_ = 0.102, indicating different developmental patterns across academic years between males and females. The main effect of academic year alone was not significant, F(3, 2349) = 1.37, *p* = 0.250, η^2^_p_ = 0.004.

### 3.3. Social Networking

Significant main effects were found for academic year, F(3, 2349) = 15.87, *p* < 0.001, η^2^_p_ = 0.02, and sex, F(1, 2349) = 8.36, *p* = 0.004, η^2^_p_ = 0.004. Additionally, the interaction between academic year and sex was significant, F(3, 2349) = 4.45, *p* = 0.004, η^2^_p_ = 0.006, showing that female students exhibited a marked increase in social networking use across academic years, surpassing male students, particularly at higher academic levels.

### 3.4. Messaging

Messaging showed a significant increase across academic years, F(3, 2349) = 6.78, *p* < 0.001, η^2^_p_ = 0.009. No significant differences were found by sex, F(1, 2349) = 0.19, *p* = 0.661, η^2^_p_ < 0.001 nor was there a significant interaction, F(3, 2349) = 1.29, *p* = 0.275, η^2^_p_ = 0.002. Messaging use increased similarly across academic years for both male and female students.

Figure 1 illustrates mean scores for digital media use across academic years, separated by sex, demonstrating trends for each domain.

### 3.5. Measurement Invariance Across Sex

Before conducting the measurement invariance tests, confirmatory factor analyses (CFA) were carried out to verify the adequacy of the measurement models for both instruments used in the study. The analyses showed good model fit for both the Questionnaire of Maladaptive Behavior toward ICT and the Questionnaire on Negative Consequences of ICT, with all indices falling within acceptable thresholds. These results support the validity of the measurement models across different subscales (see Table 4).

Measurement invariance was tested sequentially (configural, metric, and scalar invariance) for each SEM by sex. All baseline configural models showed good fit (CFI ≥ 0.951, TLI ≥ 0.943, RMSEA ≤ 0.047, SRMR ≤ 0.044). Metric and scalar invariance constraints yielded statistically significant χ^2^ differences, but changes in incremental fit indices were minimal (ΔCFI ≤ 0.006, ΔRMSEA ≤ 0.002, ΔSRMR ≤ 0.003), indicating full metric and scalar invariance across sexes (Table 5 and Table 6).

### 3.6. Measurement Invariance Across Academic Years

Measurement invariance analyses across academic years also demonstrated good fit indices at the configural level (CFI ≥ 0.939, TLI ≥ 0.929, RMSEA ≤ 0.047, SRMR ≤ 0.044). Constraining factor loadings (metric invariance) and intercepts (scalar invariance) resulted in statistically significant χ^2^ differences, yet again incremental fit indices showed negligible changes (ΔCFI ≤ 0.005, ΔRMSEA ≤ 0.001, ΔSRMR ≤ 0.003), supporting full metric and scalar invariance across academic-year cohorts (Table 7 and Table 8). Detailed structural equation models, including standardized regression weights for each latent variable and their indicators across sex and academic year, can be consulted in the Appendix A. 

Overall, these results confirm domain-specific patterns of digital media engagement, clear sex-related differences (particularly in video games and social networking), and developmental trends. Additionally, the robustness of measurement and structural models across demographic groups validates the generalizability of the relationships among emotion regulation, problematic digital media use, and negative outcomes.

## 4. Discussion

The present study examined whether the associations between emotional regulation (ER) difficulties, compulsive digital media use, cognitive preoccupation, and negative consequences remain invariant across sex and academic year in adolescents. The results confirmed metric and scalar invariance in all models, indicating that the underlying psychological processes operate similarly across these demographic groups. This is a key finding, as it demonstrates that the mechanisms linking emotional dysregulation and problematic digital media use are stable and generalizable throughout adolescence. Invariance ensures that the constructs assessed are psychometrically equivalent, validating the use of a single structural model across diverse subpopulations ([4]; [5]). While previous research has often explored sex differences in the prevalence or intensity of digital media use ([2]), our results highlight that the underlying psychological structure of risk is consistent across groups.

Consistent with prior findings, difficulties in both negative and positive emotion regulation significantly predicted adverse outcomes across all digital media domains. This supports [8] ([8]) theoretical model of emotion regulation and aligns with meta-analytic evidence linking ER difficulties to maladaptive behaviors ([1]). Furthermore, compulsive use and cognitive preoccupation emerged as strong predictors of negative outcomes, replicating previous work connecting impulsivity and rumination with problematic internet and gaming behaviors ([12]; [10]; [3]).

Although some earlier studies have suggested demographic differences in digital media use ([9]), our findings demonstrate that both the measurement and structural relationships remain consistent across sex and academic years. These results suggest the existence of a consistent psychological pathway—linking ER difficulties and problematic usage to negative outcomes—across sex and academic year groups in our adolescent sample.

### 4.1. Clinical and Research Implications

These results have important implications for intervention design. The demonstrated invariance suggests that universal strategies—such as school-based programs focused on strengthening emotion regulation and reducing compulsive media use—can be implemented effectively without tailoring by sex or age. This is supported by longitudinal studies showing similar developmental trajectories for problematic digital media use among boys and girls ([6]), and by neurodevelopmental evidence indicating that, despite structural brain differences, adolescents of both sexes follow comparable patterns in emotional maturation ([18]). 

### 4.2. Limitations

Several limitations should be acknowledged. First, the cross-sectional design limits causal inference and prevents assessment of the temporal stability of the observed relationships. Longitudinal designs are needed to explore these dynamics more thoroughly. Second, reliance on self-report measures may introduce bias due to shared method variance, social desirability, or recall errors. Future research should consider incorporating objective behavioral metrics or multi-informant methods. Third, although the sample size (N = 2357) and its diversity in educational settings enhance representativeness, the generalizability beyond adolescents in the Madrid region remains uncertain. Additionally, test–retest reliability was not assessed, and other potentially influential demographic moderators—such as socioeconomic status or cultural background—were not included in the analysis.

## Figures and Tables

**Figure 1 ejihpe-15-00145-f001:**
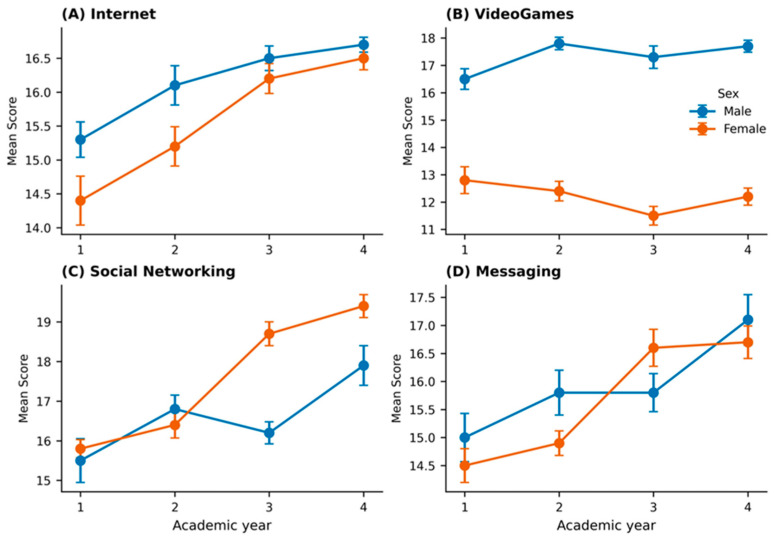
Mean self-reported usage scores across academic years for (**A**) Internet, (**B**) Video Games, (**C**) Social Networking, and (**D**) Messaging, stratified by sex. Error bars represent standard errors of the mean (±SEM). Male participants are represented in blue and female participants in orange. A progressive increase in usage across academic years is observed for most domains, with notable sex differences particularly in Video Games and Social Networking.

**Table 1 ejihpe-15-00145-t001:** Distribution of participants across academic years by sex and age. Values are presented as number of participants with percentages in parentheses for each subgroup. Age is expressed as mean and standard deviation (M ± SD). Percentages may not add up to exactly 100% due to rounding.

Academic Year	Total N (%)	Male N (%)	Female N (%)	Male Age (M ± SD)	Female Age (M ± SD)
1º	720 (30.5)	375 (29.5)	345 (31.7)	12.30 (0.50)	12.31 (0.52)
2º	694 (29.4)	370 (29.1)	324 (29.8)	13.38 (0.58)	13.29 (0.50)
3º	515 (21.8)	275 (21.7)	240 (22.1)	14.40 (0.61)	14.32 (0.50)
4º	428 (18.2)	250 (19.7)	178 (16.4)	15.44 (0.65)	15.79 (0.57)
	2357 (100)	1270 (53.9)	1087 (46.1)		

**Table 2 ejihpe-15-00145-t002:** Descriptive statistics of test scores.

	Mean	Std. Deviation
Positive mood regulation	8.74	4.48
Negative emotion regulation	7.05	3.42
Compulsive Use	8.24	3.69
Cognitive preoccupation	7.03	3.39
Internet	15.78	6.13
Video Games	14.98	7.67
Social Networking	16.88	7.91
Messaging	15.69	7.63

**Table 3 ejihpe-15-00145-t003:** Pearson’s correlation matrix between the study variables.

	Positive Mood Regulation	Negative Emotion Regulation	Compulsive Use	Cognitive Preoccupation
Internet	0.440 **	0.295 **	0.481 **	0.083 **
Video Games	0.391 **	0.289 **	0.305 **	0.084 **
Social Networking	0.430 **	0.299 **	0.454 **	0.146 **
Messaging	0.455 **	0.293 **	0.363 **	0.138 **

Note: 2357 participants were evaluated. ** *p* < 0.01.

**Table 4 ejihpe-15-00145-t004:** Confirmatory factor analysis of the measured variables.

Models	χ^2^	CFI	TLI	RMSEA	SRMR
Questionnaire of Maladaptive Behavior toward ICT	387.398	0.970	0.962	0.043	0.032
Questionnaire on Negative Consequences of ICT					
Internet browsing	78.946	0.981	0.971	0.044	0.023
Video games	130.397	0.982	0.973	0.059	0.022
Social networking	134.441	0.982	0.974	0.060	0.022
Messaging	121.441	0.986	0.979	0.057	0.020

χ^2^: Chi-square; CFI (Robust Comparative Fit Index); TLI (Tucker–Lewis Index); RMSEA (Root Mean Square Error of Approximation); SRMR (Standardized Root Mean-square Residual). Note: In all models, the degrees of freedom were 72 for the Questionnaire of Maladaptive Behavior toward ICT, 14 for all models of Questionnaire on Negative Consequences of ICT. The significance level of the fit indices for all models and analyses was *p* < 0.001.

**Table 5 ejihpe-15-00145-t005:** Fit indices for all the models, multigroup analysis using sex.

Models	χ^2^	CFI	TLI	RMSEA	SRMR
Internet					
Configural invariance	1098.765	0.951	0.943	0.042	0.037
Metric invariance	1157.046	0.949	0.942	0.042	0.040
Scalar invariance	1200.290	0.947	0.943	0.042	0.041
Videogames					
Configural invariance	1005.38	0.964	0.958	0.039	0.034
Metric invariance	1033.23	0.964	0.959	0.039	0.035
Scalar invariance	1152.10	0.958	0.955	0.041	0.037
Social Networking					
Configural invariance	1125.10	0.959	0.952	0.043	0.034
Metric invariance	1156.47	0.958	0.953	0.042	0.036
Scalar invariance	1250.62	0.954	0.950	0.043	0.037
Messaging					
Configural invariance	988.81	0.967	0.962	0.039	0.034
Metric invariance	1018.16	0.967	0.963	0.038	0.035
Scalar invariance	1074.70	0.965	0.962	0.039	0.036

n all models, the degrees of freedom were 358 for the configurational analysis, 374 for the metric analysis, and 390 for the scalar analysis. The significance level of the fit indices for all models and analyses was *p* < 0.001.

**Table 6 ejihpe-15-00145-t006:** Metric and scalar invariance analysis summary. Increases in fit indices for the four models by sex.

Models	Δχ^2^	*p*-Value	ΔCFI	ΔTLI	ΔRMSEA	ΔSRMR
Internet						
Metric invariance	58.282	0.000	−0.002	−0.001	0.000	0.003
Scalar invariance	43.244	0.000	−0.002	0.001	0.000	0.001
Videogames						
Metric invariance	27.85	0.033	0.000	0.001	0.000	0.001
Scalar invariance	118.87	0.000	−0.006	−0.004	0.002	0.002
Social Networking						
Metric invariance	31.37	0.012	−0.001	0.001	−0.001	−0.001
Scalar invariance	94.15	0.000	−0.004	−0.003	0.001	0.001
Messaging						
Metric invariance	29.35	0.022	0.000	0.001	−0.001	0.001
Scalar invariance	56.54	0.000	−0.002	−0.001	0.001	0.001

Δχ^2^: Invariance analysis. In all models, the difference in degrees of freedom was 16.

**Table 7 ejihpe-15-00145-t007:** Fit indices for all the models, multigroup analysis using academic year.

Models	χ^2^	CFI	TLI	RMSEA	SRMR
Internet					
Configural invariance	1648.61	0.939	0.929	0.047	0.044
Metric invariance	1713.791	0.938	0.932	0.046	0.046
Scalar invariance	1848.245	0.933	0.930	0.047	0.048
Video games					
Configural invariance	1498.478	0.958	0.951	0.043	0.038
Metric invariance	1570.219	0.957	0.953	0.042	0.041
Scalar invariance	1661.337	0.955	0.953	0.042	0.042
Social Networking					
Configural invariance	1616.883	0.952	0.944	0.046	0.040
Metric invariance	1671.740	0.952	0.947	0.045	0.042
Scalar invariance	1785.949	0.948	0.946	0.045	0.044
Messaging					
Configural invariance	1467.148	0.961	0.955	0.042	0.039
Metric invariance	1535.684	0.960	0.956	0.041	0.042
Scalar invariance	1637.768	0.958	0.956	0.042	0.043

In all models, the degrees of freedom were 716 for the configurational analysis, 764 for the metric analysis, and 812 for the scalar analysis. The significance level of the fit indices for all models and analyses was *p* < 0.001.

**Table 8 ejihpe-15-00145-t008:** Metric and scalar invariance analysis summary. Increases in fit indices for the four models by academic year.

Models	Δχ^2^	*p*-Value	ΔCFI	ΔTLI	ΔRMSEA	ΔSRMR
Internet						
Metric invariance	65.183	0.049	−0.001	0.003	−0.001	0.002
Scalar invariance	134.45	0.000	−0.005	−0.002	0.001	0.002
Video games						
Metric invariance	71.74	0.014	−0.001	0.002	−0.001	0.003
Scalar invariance	91.11	0.000	−0.002	0.000	0.000	0.001
Social Networking						
Metric invariance	54.85	0.230	0.000	0.003	−0.001	0.002
Scalar invariance	114.20	0.000	−0.004	−0.001	0.000	0.002
Messaging						
Metric invariance	68.53	0.028	−0.001	−0.001	−0.001	0.003
Scalar invariance	102.08	0.000	−0.002	0.001	0.001	0.001

Δχ^2^: Invariance analysis. In all models, the difference in degrees of freedom was 48.

## Data Availability

The raw data supporting the conclusions of this article will be made available by the authors on request.

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
