# Peer review of "Problematic Internet Use: Measurement and Structural Invariance Across Sex and Academic Year Cohorts"

_ejihpe, 2025, doi:10.3390/ejihpe15080145_

Round 1
Reviewer 1 Report
Comments and Suggestions for Authors
The article "Problematic Internet Use: Measurement and Structural Invariance Across Sex and Academic Year Cohorts" deals with a relevant topic of problematic internet use among adolescents. In my opinion, the strong point of the article is the relatively clear definition between different types of internet use, as problematic internet users are different in their behavioral patterns and can struggle with different problems based on their preferences. The article also clearly aims for validation or further development of the author's unpublished questionnaires, which can probably become useful psychodiagnostic tools in the future. It probably explains why the article is mainly focused on the results of structural equation modeling, which was used adequately, to me.
However, the article, in my opinion, requires some broader explanation for the methodological part. First of all, among the two questionnaires used, the Questionnaire of Maladaptive Behavior toward ICT is currently unpublished (as stated in the reference list), and the Questionnaire on Negative Consequences of ICT is referenced as published in a conference paper in Spanish. Due to this, I suggest adding more information about the questionnaires used in the article. The article includes the information about the number of items in the first questionnaire, but none about the second one. It is not stated how many items form each subscale, which is relevant to understanding the relatively low (below 0.8) Cronbach's alphas for most subscales - with less items in a subscale lower alphas are less concerning due to how this statistics works. The inclusion of subscale parameters will also help in understanding the mean scores in Figure 1, as in the current state of the article, we have no idea about the min and max possible scores for each subscale. It would probably be useful if the authors expanded the information on the subscales a bit with the examples of the items or some other description of their understanding of problematic internet use symptoms and, e.g., how the Questionnaire on Negative Consequences of ICT distinguishes between social networking and messaging (as most social networks also provide access to their specific messaging tools as well). It is also unclear if the negative consequences include the same problems for each type of online activite or if those consequencies are domain specific.
My second main question is also related to the methodological part of the article. First of all, it is not clearly explained anywhere in the article why the authors aimed for academic year cohorts instead of the age of the participants. Additionally, if the mean ages and standard deviations are presented correctly in Table 1, some clarification about the sample might be relevant. I am not very familiar with compulsory secondary education in Spain, but according to the Internet, students study there from 12 to 16 years old, with the possibility to repeat 1 year in the first half and 1 year in the second half of the education, but not the last year. If this is correct, age distribution for some groups looks odd. E.g., the first-year female subsample has M (± SD) described as 12.42 (2.09), but there are probably little to no 10-year-old students in the subgroup? For the eldest female group (year 4), SD is as high as 3.51, meaning that there are people aged from 12.28 y.o. (which is younger than the mean age in year 1) to 19.3 years old? Is this a regular sample, or are there any demographic specifics to take into consideration? If the sample had just a few significant outliers leading to such a strange distribution, why were they not excluded? Still, the question remains, why the academic year cohorts and not the age were used as a parameter in the models, while age and its possible contribution were mostly ignored. Anyway, the article would benefit from the authors further explaining why they considered the academic year cohort better for their study design compared to the age of the students.
The last sentence in the discussion section concludes that "This suggests a shared psychological pathway through which ER difficulties and problematic usage contribute to adverse consequences, irrespective of demographic group"; however, the only demographic factors in the study are biological sex, academic year, and age, the latter stated but not used in the models. I do not think it is possible to make a statement about demographic groups in this case, as the study says nothing about the income, ethnicity, or religion of the participants, all of which can contribute to internet use behaviors. I suggest the authors should make statements within the data they have obtained without too broad generalization.
Otherwise the article is interesting and relevant but is more focused on SEM involving the specific non-commonly known questionnaires than anything else, which might make it less interesting for the reader unfamiliar with the very specific questionnaires used. Overall, I suggest expanding it in the methodological and discussion parts to make the readers more invested in the text.
While the text seems good from the grammatical point of view, additional proofreading won't be harmful. At least I seemingly spotted an unneeded comma in the abstract section that makes a research description a bit confusing.
Reviewer 2 Report
Comments and Suggestions for Authors
Thanks for the opportunity to review this thoughtful and potentially valuable manuscript. I have some suggestions for the author team to consider when improving their manuscript.
First, the key term "problematic digital media use" should be kept consistent throughout the manuscript. In the current manuscript, this key term has been changing frequently, from problematic media use to problematic media use, which can lead to unnecessary confusion and ambiguity.
Second, the manuscript aimed to demonstrate the negative consequences of problematic digital use, but I did not find such analyses and results. The authors should conduct regression analyses to examine the associations of four dimensions of problematic digital media use and negative consequences.
Third, the measurement invariance tests should be conducted for the problematic digital media use rather than for each dimension of the negative consequences of problematic digital media use (i.e., internet, online borrowing, video games, and social networking).
Fourth, before the measurement invariance tests, the authors should conduct CFA on the problematic digital media use, and the negative consequences of problematic digital media use, respectively.
Fifth, the authors are suggested to report the contents of each test including the content of each item.
Sixth, when reporting the results of ANOVA, the authors should also report the effect sizes (e.g., eta squared).
Seventh, I suggest that the authors add a table of descriptive statistics and correlations among the study variables.
Finally, I think that a native English speaker should help the author team to polish and improve the language.
